

# Hygroscopic Growth and Activation Changed Submicron Aerosol Composition and Properties in North China Plain

Weiqi Xu[1], Ye Kuang[2], Wanyun Xu[3], Zhiqiang Zhang[1,4], Biao Luo[2], Xiaoyi Zhang[3,5], Jiangchuang Tao[2], Hongqin Qiao[2], Li Liu[6] and Yele Sun[1,4*]

[1]State Key Laboratory of Atmospheric Boundary Layer Physics and Atmospheric Chemistry, Institute of Atmospheric Physics, Chinese Academy of Sciences, Beijing 100029, China.

[2]Institute for Environmental and Climate Research, Jinan University, Guangzhou 511443, China

[3]State Key Laboratory of Severe Weather & Key Laboratory for Atmospheric Chemistry, Institute of Atmospheric Composition, Chinese Academy of Meteorological Sciences, Beijing 100081, China

[4]College of Earth and Planetary Sciences, University of Chinese Academy of Sciences, Beijing 100049, China

[5]Department of Atmospheric and Oceanic Sciences, Fudan University, Shanghai 200438, China

[6]Key Laboratory of Regional Numerical Weather Prediction, Institute of Tropical and Marine Meteorology, China Meteorological Administration, Guangzhou 510080, China

*Corresponding author:* Yele Sun (sunyele@mail.iap.ac.cn) and Ye Kuang (kuangye@jnu.edu.cn)

**Abstract.** Aerosol hygroscopic growth and activation under high relative humidity (RH) conditions significantly influence the physicochemical properties of submicron aerosols ($PM_1$). However, this process remains poorly characterized due to limited measurements. To address this gap, we deployed an advanced aerosol-fog sampling system that automatically switched between $PM_1$, $PM_{2.5}$, and TSP inlets at a rural site in the North China Plain in cold season. The results revealed that aerosol swelling due to water vapor uptake influenced aerosol sampling under high RH conditions by shifting the cut-off size of impactors. Under subsaturated high RH (> 90%), over 25% of aerosol mass with dry diameters below 1 μm resided in supermicron ranges, while in supersaturated foggy conditions, more than 70% submicron aerosol migrated to supermicron ranges. Hygroscopic growth and activation particularly affected highly hydrophilic inorganic salts shifting a significant mass submicron sulfate and nitrate particles to supermicron ranges, with 27 – 33% under 95% ≤ RH ≤ 99%, and 65.5% in supersaturated foggy conditions. Moreover, more than 10% of submicron biomass burning organic aerosols grew beyond 2.5 μm during fog events, while fossil fuel-related OA (FFOA) remained dominantly in submicron ranges, suggesting inefficient aqueous conversion of FFOA. The two SOA factors (OOA1 and OOA2) behaved differently under supersaturated conditions, with OOA2 exhibiting a higher activated fraction despite a lower oxygen-to-carbon ratio. A substantial increase in organic nitrate and organosulfur mass concentrations in activated droplets during fog events suggested aqueous conversions and formations of brown carbon with potential radiative impacts. Overall, our study highlights remarkably different aqueous processing of primary and secondary $PM_1$ aerosol components under distinct ambient RH conditions.

## 1 Introduction

Submicron aerosols ($PM_1$, particulate matter with aerodynamic diameter less than 1 μm) are significant components of atmospheric aerosol particles, impacting air quality, climate change, and human health (Fuzzi et al., 2015; Shrivastava et al., 2017; Pope et al., 2002; Molina and Molina, 2004). Extensive research has been devoted to explore their



characteristics, such as mass concentrations, sources, and size distributions across diverse atmospheric conditions (Sun et al., 2015; Zhou et al., 2020; Wang et al., 2019; Zhang et al., 2011). Recent studies have also examined the influence of relative humidity (RH) on $PM_1$ physicochemical properties (Li et al., 2019; Huang et al., 2020b; Sun et al., 2013). However, these measurements were constrained by the occurrence of high RH conditions or limitations in sampling

techniques under both subsaturated and supersaturated conditions, making it challenging to accurately quantify the effects of aerosol swelling due to hygroscopic growth and activation on $PM_1$ properties. Therefore, a comprehensive understanding of the changes in size-resolved mass concentrations, composition, and properties of $PM_1$ under subsaturation and supersaturation conditions is needed.

While previous studies have provided a relatively clear understanding of the impact of RH on the mass concentrations

of secondary inorganic aerosols (SIA), our comprehension of its effects on the evolution of organic aerosol (OA) species remains incomplete. For example, coal combustion OA (CCOA) increases with rising RH in Beijing during wintertime (Sun et al., 2013), whereas it remains relatively stables as a function of RH in Xi'an (Elser et al., 2016). Furthermore, Gilardoni et al. (2016) demonstrated that the aqueous-phase processing of emissions from biomass burning contributes significantly to the formation of secondary organic aerosols (SOA), while Duan et al. (2022) emphasized the predominant

role of photochemical oxidation in comparison to aqueous-phase processing in the contributions of biomass burning emissions to SOA. In addition, many studies have suggested that aqueous processing plays a significant role in changing SOA physicochemical properties, and the influences depend strongly on SOA species. For instance, Hu et al. (2016) proposed that aqueous chemistry could enhance the formation of less oxidized SOA (LO-OOA), while Xu et al. (2017) found the largely increased more oxidized SOA (MO-OOA) with the increase of RH in Beijing across three seasons.

Furthermore, the changes in organic aerosol oxidation, often indicated by $f_{44}$ (fraction of $m/z$ 44 in OA) and oxygen-to-carbon (O/C) ratio, as a function of RH exhibit noticeable discrepancies in different environments. Wang et al. (2016) observed an increase in O/C ratio at high RH levels, while Sun et al. (2013) reported a significant decrease in $f_{44}$ followed by minor variations at high RH levels. Huang et al. (2020a) also demonstrated a decrease in average carbon oxidation state ($OS_C = 2 \times O/C - H/C$) with increasing liquid water content. Besides the differences in precursors and chemical

processes, the enlargement of particle size due to water condensation beyond the sampling cut-off size may also contribute to these discrepancies. However, the studies regarding the effect of particle sizes on OA composition and oxidation degree remain limited.

Numerous studies have observed a shift in the peak diameter of PM species towards larger sizes during fog periods, with some particles growing into the supermicron range (Ge et al., 2012; Chen et al., 2022; Gupta and Elumalai, 2018). For

instance, Wang et al. (2015) documented a notable increase in the $PM_{1-2.5}$ fraction in Beijing during highly polluted days, while Elser et al. (2016) reported significant enhancements of SIA components and SOA in sizes over 1 μm during extreme haze episodes in Beijing and Xi'an. Additionally, Chen et al. (2018) highlighted that aerosol hygroscopic growth could induce a shift in the size distribution of dry state aerosols as captured by impactors. Moreover, recent studies have conducted simultaneous comparisons of the physicochemical properties of $PM_1$ and $PM_{2.5}$ (Li et al., 2023; Sun et al.,

2020; Zheng et al., 2023), investigating chemical disparities caused by different particle sizes. Sun et al. (2020) observed different RH dependence of $PM_1/PM_{2.5}$ species due to different hygroscopicity and phase states. In addition, Zheng et al.



(2023) noted a larger increase in MO-OOA with diameters ranging from 1 to 2.5 μm compared to LO-OOA under conditions of high aerosol liquid water content. Despite their significant implications for the variation of aerosol mass loading and chemical composition caused by size differences, a direct quantification of the contribution of hygroscopic growth and activation of PM species has not yet been achieved.

In this study, we conducted real-time measurements of size-resolved $PM_1$ species using a High-Resolution Aerosol Mass Spectrometer (HR-AMS), an Aerodynamic Aerosol Classifier (AAC) in tandem with a Condensation Particle Counter (CPC), combined with an advanced aerosol-fog sampling system that is capable of switching automatically between inlets with three different cut-off sizes. This allowed us to investigate how aerosol hygroscopic growth and activation impacted the size shifts of various $PM_1$ components under various ambient RH conditions. The impacts of RH, particularly high RH levels, on changes in mass concentrations and size distributions of PM species, as well as the oxidation degree of OA under different sizes were assessed.

## 2 Materials and methods

### 2.1 Sampling and data analysis

Field campaigns were conducted in Gucheng, a representative rural site in NCP, from 11 October to 14 November 2021. Ambient particles and droplets were initially selected using an advanced aerosol-cloud sampling inlet system, which alternated between the $PM_1$ cyclone, $PM_{2.5}$ cyclone, and TSP passage every 20 minutes. The filtered particles were then dried using a Nafion dryer to maintain inlet RH below 20% throughout the campaigns. Size-resolved submicron aerosols species, including organics (Org), sulfate ($SO_4$), nitrate ($NO_3$), ammonium ($NH_4$), and chloride (Chl), were measured using a HR-AMS. Note that the $PM_1$, $PM_{2.5}$ and TSP species in this study refer to the PM species measured by HR-AMS, which is selected using the $PM_1$ cyclone, $PM_{2.5}$ cyclone, and TSP passage. Additionally, an AAC operated in tandem with a CPC employing a sheath-to-sampling flow ratio of 10 facilitated measurements of aerosol size distributions spanning aerodynamic diameters from approximately 200 nm to 4 μm. A more detailed description of the sampling procedures and used instruments can be found in Kuang et al. (2024).

The ionization efficiency (IE) was calibrated using pure $NH_4NO_3$ particles with a diameter of 300 nm following standard protocols (Jayne et al., 2000). The relative ionization efficiency (RIE) of ammonium and sulfate were determined as 5.3 and 1.3, respectively, while the default RIE values were utilized for organic species (1.4), nitrate (1.1), and chloride (1.3). Composition-dependent collection efficiency (Middlebrook et al., 2012) was employed in this study. Elemental ratios were derived using the "Improved-Ambient (I-A)" method (Canagaratna et al., 2015), which includes calculations for O/C, hydrogen-to-carbon (H/C), nitrogen-to-carbon (N/C), and organic mass-to-organic carbon (OM/OC) ratios.

### 2.2 Source apportionment of OA

Two primary organic aerosol (POA) factors, namely biomass burning OA (BBOA) and fossil fuel-related OA (FFOA), along with two SOA factors, were identified using positive matrix factorization (PMF) (Ulbrich et al., 2009; Paatero and



Tapper, 1994). The mass spectra of OA factors and correlations between these factors and external species are presented
in Figs. S1-S2. The BBOA spectrum was characterized by prominent $m/z$ 60 (mainly $C_2H_4O_2^+$, $f_{60}$=1.8%) and 73 (mainly
$C_3H_5O_2^+$, $f_{73}$=1.1%), which are well-established maker ions of biomass burning (Mohr et al., 2009). Additionally, BBOA
showed strong correlations with $C_2H_4O_2^+$ ($R^2$=0.96) and $C_3H_5O_2^+$ ($R^2$=0.96). Consistent with previous studies in Beijing
(Xu et al., 2019), a mixed factor termed FFOA, comprising contributions from traffic emissions and coal combustion,
was identified by PMF. FFOA was characterized by typical hydrocarbon ion series, yet a relatively high $f_{44}$ (0.089) was
observed, likely from the aging during regional transport, consistent with observations in winter 2016 in Beijing (Xu et
al., 2019) and the previous study in Gucheng (Chen et al., 2022). Although the mass spectra of the two SOA factors both
showed high $f_{44}$ (0.16-0.21) and O/C (0.78-0.91), they showed different spectral patterns, correlation with tracers, and
diurnal variations, suggesting different chemical processing and formation mechanisms for these two factors. For
instance, OOA1 exhibits a pronounced increase during daytime (Fig. S3), whereas OOA2 remains relatively constant
throughout the day. Furthermore, OOA1 exhibits higher $CO_2^+/C_2H_3O^+$ (3.9) and O/C (0.91) values compared to OOA2
($CO_2^+/C_2H_3O^+$ =2.1, O/C=0.78). The significant signal of $f_{29}$ (0.076) and high N/C (0.014) were observed in OOA2
compared to OOA1, suggesting that OOA2 is more like a factor associated with aqueous-phase processing (Zhao et al.,
2019).

## 3. Results and discussion

### 3.1 General descriptions



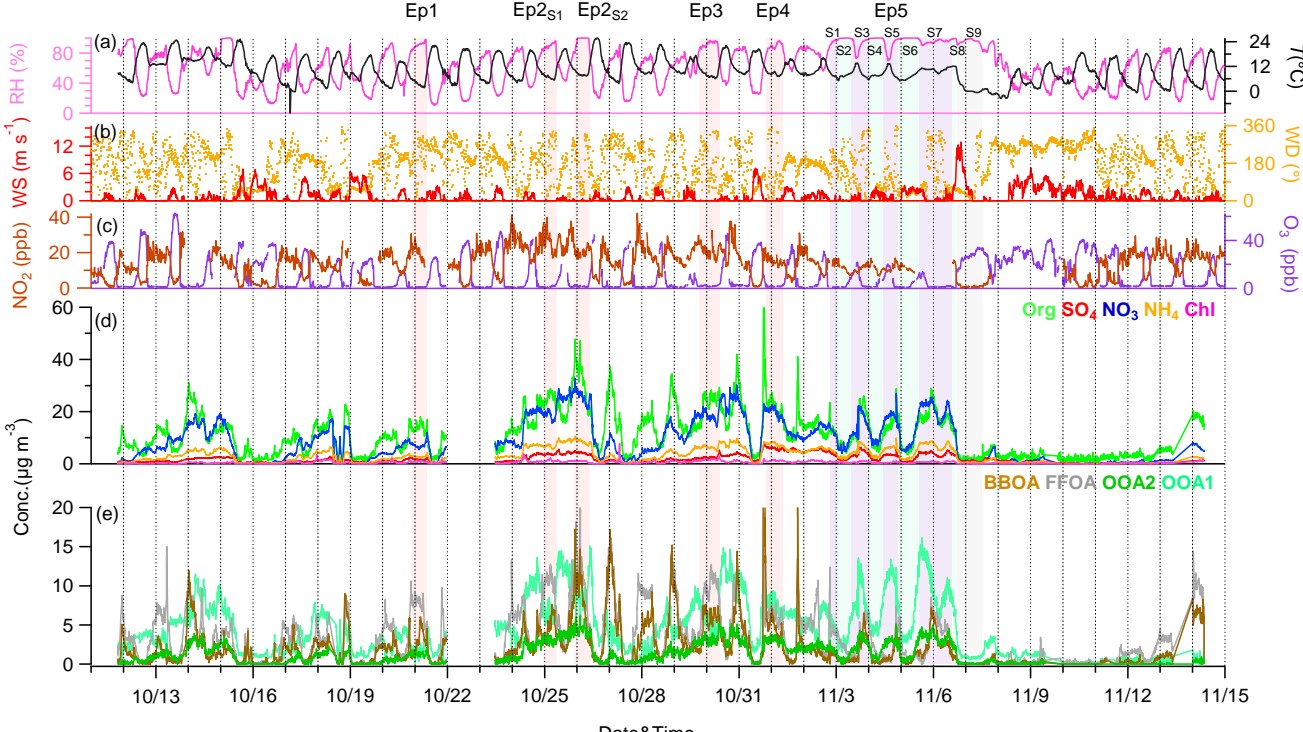

**Figure 1**. Time series of meteorological variables, gaseous species, and PM$_1$ species. In addition, five episodes (Ep1-5) with relatively high RH levels (> 95%) are marked for further discussion, with Ep5 being a 3-day consecutive fog event that was further divided into nine stages (S1-S9).

Figure 1 illustrates the temporal variations of meteorological parameters, gaseous pollutants, and PM$_1$ species for the entire study. Overall, the mass concentrations of PM species under different sizes were comparable (Fig. S4), except during episodes with high RH levels (> 95%). Five episodes (Ep1-Ep5 in Fig. 1) with high RH levels exhibited notable differences in the properties of PM species under different sizes. Further details are provided in Sect. 3.2-3.5. These results suggest significant impacts of aerosol hygroscopic growth and activation on aerosol sampling under high RH and supersaturated conditions. Unless otherwise specified, the characterizations of aerosols presented in this section refer to PM$_1$ species. For reference, the mass concentrations of PM$_{2.5}$ and TSP species are also provided in Table S1. The mass concentrations of NR-PM$_1$ varied from 0.58 μg m$^{-3}$ to 123.7 μg m$^{-3}$, with an average of 25.4 μg m$^{-3}$. Compared to the measurements in winter 2018 and 2019 at the same site, a notable decrease in OA mass concentration was observed (36.6 μg m$^{-3}$ in winter 2018 vs. 11.2 μg m$^{-3}$ in winter 2021), while the mass concentrations of SO$_4$ and NO$_3$ initially increased in 2019 and then decreased to 1.8 μg m$^{-3}$ and 8.7 μg m$^{-3}$, respectively, in winter 2021 (Fig. S5). These changes are strongly associated with the largely decreased coal combustion emission in rural areas in NCP, and the varying responses of volatile organic compounds (VOCs) and SO$_2$/NO$_2$ to the emission changes. Additionally, the fraction of NO$_3$ exhibited a continuous increase from 2018 (17%) to 2021 (34%), leading to an increase in the NO$_3$/SO$_4$ ratio from 1.4 in winter 2018 to 4.8 in winter 2021, highlighting the increasing importance of nitrate in the NCP (Lei et al., 2021).





In comparison to 2018, the decrease in POA concentration in 2021 (23.0 μg m$^{-3}$ vs. 4.8 μg m$^{-3}$) was more significant than that of SOA (13.5 μg m$^{-3}$ vs. 6.1 μg m$^{-3}$), indicating the direct impact of pollution control policies on primary emissions in NCP (Liu et al., 2023; Hu et al., 2023). Consequently, the fraction of SOA increased from 35.9% in winter 2018 to 50.2% in winter 2021, highlighting the increased oxidation of organic aerosols. This increase in oxidation is further supported by the rise in the O/C ratio from 0.51 in 2019 to 0.55 in 2021. Notably, the O/C observed in Gucheng was higher than those at urban sites with an average O/C ratio of 0.43 in Asia (Zhou et al., 2020), suggesting a greater oxidation of OA in suburban areas. Additionally, the frequent occurrence of episodes with relatively high RH levels, which were characterized by relatively high O/C ratios (0.44-0.67 in Ep1-Ep5 on average) compared to urban sites, could contribute to this difference.

The size distributions of PM species also exhibited variations at the same sampling site when compared to previous winter. The peak diameters of SO$_4$ and NO$_3$ were found to be similar (~500 nm, Fig. S6), contrasting with the results observed in winter 2019 in Gucheng, where SO$_4$ peaked at ~600 nm and NO$_3$ at ~400 nm (Chen et al., 2022). Additionally, a larger peak diameter of OA (450 nm) was observed in winter 2021 compared to winter 2019 (400 nm). The discrepancy in size is likely attributed to the more frequent occurrence of high RH events in winter 2019 (Fig. S7), leading to variations in aqueous-phase processes and hygroscopic growth. Notably, the differences in size distributions of SO$_4$ and NO$_3$ between PM$_{2.5}$ and TSP are negligible, whereas smaller peak diameters were observed for PM$_1$ species. Such results indicate that hygroscopic growth of aerosols ranging from 1 to 2.5 μm is relatively facile, while it becomes more challenging for them to grow beyond 2.5 μm. In contrast, OA exhibited similar peak diameters under different sampling size cutoff, although the mass concentrations of PM$_1$ OA above ~700 nm and in the range of 200-400 nm were lower compared to PM$_{2.5}$ and TSP OA. These differences in mass concentrations above 700 nm are likely attributable to contributions from SOA, while the variations in the 200-400 nm range may be influenced by the conversion of POA during aqueous-phase processes.

## 3.2 Impact of aerosol hygroscopic growth and activation on size distributions



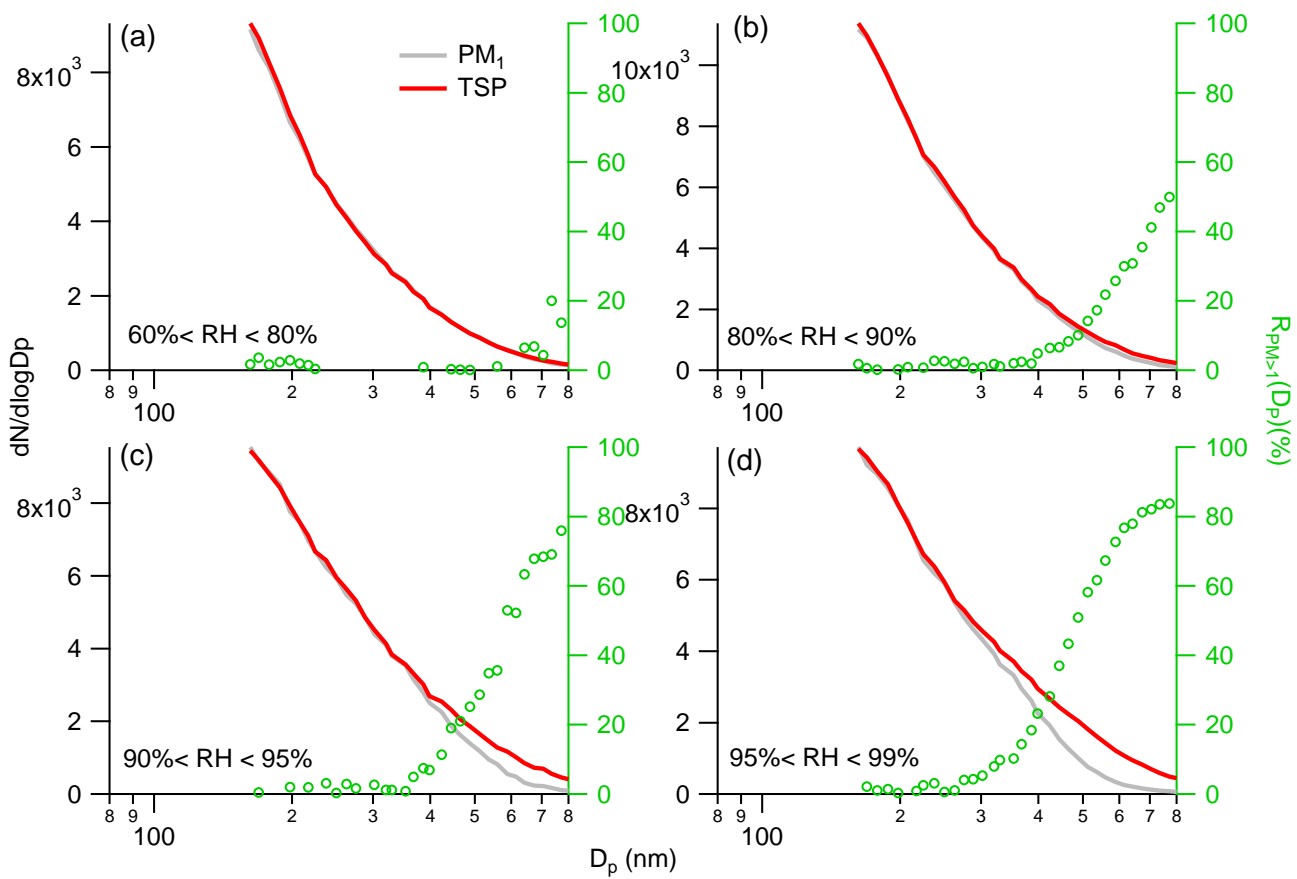

**Figure 2**. Average particle number size distributions of PM$_1$ and TSP under different RH ranges on the left axis, with the right axis shows the corresponding average $R_{PM_{>1}}$(Dp).

In this section, $R_{PM_{>1}}(D_p) = \frac{N(D_p, TSP) - N(D_p, PM_1)}{N(D_p, TSP)}$ and $R_{PM_{>2.5}}(D_p) = \frac{N(D_p, TSP) - N(D_p, PM_{2.5})}{N(D_p, TSP)}$ are defined to indicate the size-resolved ratios of submicron aerosols that grow above 1 and 2.5 μm through water condensation, respectively. Note that the aerodynamic diameter of the AAC was converted to mobility diameter assuming a spherical aerosol shape and an aerosol density of 1.6 g cm⁻³. Subsaturated (measured RH < 99%) and supersaturated conditions (determined based on aerosol activation characteristics of submicron aerosols) were individually investigated to discuss the impacts of hygroscopic growth and activation (in fogs).

We focus on comparing the size distributions of PM$_1$ and TSP, as well as the corresponding average $R_{PM_{>1}}(D_p)$ due to the negligible differences in size distributions of PM species between PM$_{2.5}$ and TSP under subsaturated conditions (Fig. S8). Note that under the conditions of low RH (< 60%), there was no activation of aerosols. In the RH range of 60-80%, there were minimal differences observed in the size distributions between PM$_1$ and TSP (Fig. 2). Only small fractions (<20%) of aerosols larger than 700 nm were found to grow beyond an aerodynamic diameter of 1 μm. As RH increased



above 95%, the minimal size for growing beyond PM$_1$ shifted to about 300 nm. A significant portion (>50%) of aerosols larger than 500 nm was observed to grow beyond an aerodynamic diameter of 1 μm when RH varied from 95 to 99%. These results suggest that at relative humidity levels above 90%, a considerable fraction (>25%) of submicron aerosol mass is present in supermicron diameter ranges, which may significantly enhance multiphase reactions of aerosols.


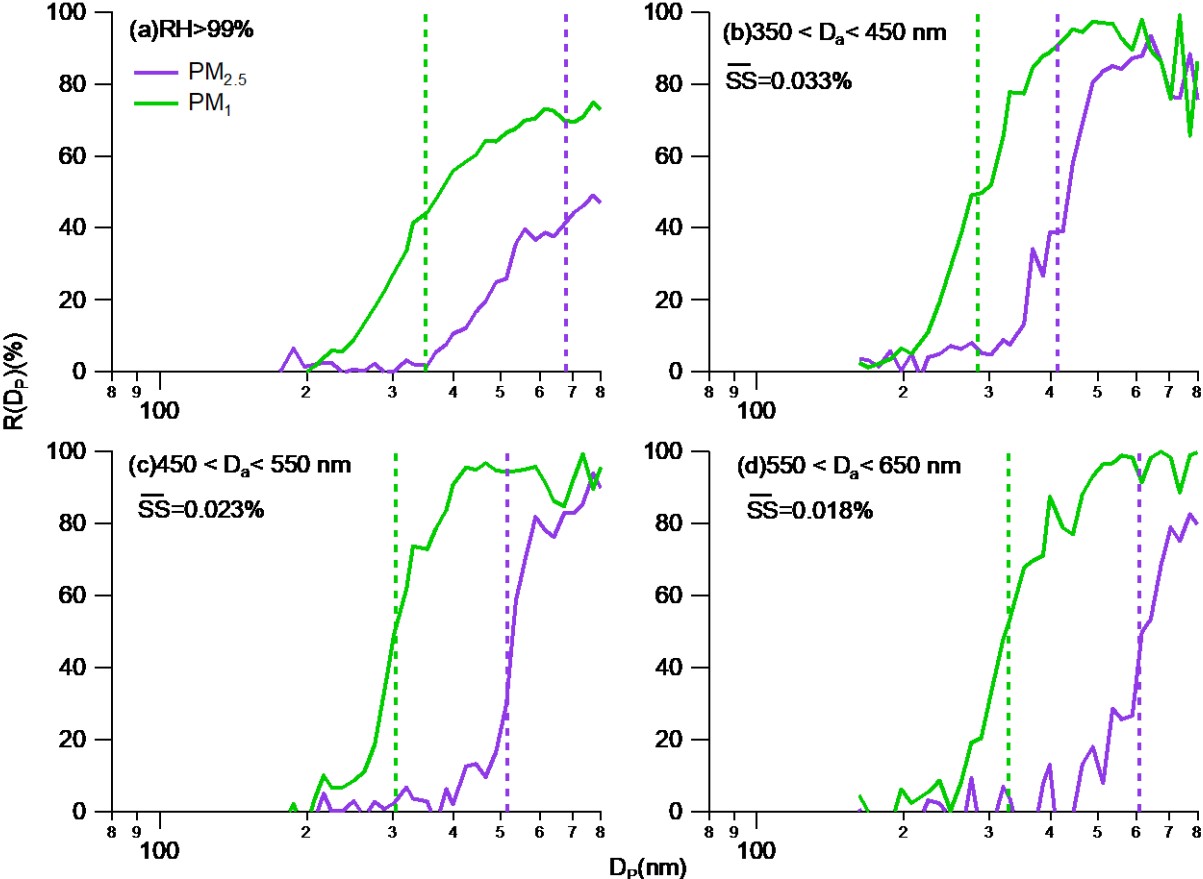

**Figure 3**. Average $R_{PM_{>1}}(D_p)$ and $R_{PM_{>2.5}}(D_p)$ as a function of D$_p$ under different critical activation diameter (D$_a$) ranges. The green and purple dash lines refer to the fitted average critical diameter that grow beyond PM$_1$ ($D_{a,PM_1}$) and PM$_{2.5}$ (D$_a$).

As relative humidity exceeded 99%, aerosol activation was typically observed most of the time, although it may not occur in every instance. The average $R_{PM_{>1}}(D_p)$ and $R_{PM_{>2.5}}(D_p)$ curves as a function of D$_p$ are presented in Fig.3, showing that the minimal size for growing beyond PM$_1$ shifted to about 200 nm and portions of dry state submicron aerosols would grow even beyond PM$_{2.5}$ through aerosol hygroscopic growth and activation. To further investigate how aerosol activation processes impacts on the ambient size shift, the size-resolved activation ratio of aerosols in fogs




represented by $R_{PM_{>2.5}}(D_p)$ and $R_{PM_{>1}}(D_p)$ were further analyzed for a case of a 3-day consecutive fog event (Ep5 in

Fig.1). Critical activation diameters ($D_a$), representing the submicron diameter at which activation beyond 2.5 µm occurs, fitted using the error function erf as detailed in Kuang et al. (2024), ranged from 362 nm to 664 nm, with an average of 487 nm. The supersaturation ratios observed during fogs, as calculated based on the aerosol activation theory proposed by Petters and Kreidenweis (2007), ranged from 0.016% to 0.042%, with an average value of 0.026%. Figure 3 illustrate the average $R_{PM_{>2.5}}(D_p)$ and $R_{PM_{>1}}(D_p)$ curves under three different $D_a$ ranges, along with the corresponding average

supersaturation ratios. Interestingly, as the average $D_a$ increased from 413 to 607 nm, the average critical diameter ($D_{a,PM_1}$) required to grow beyond 1 µm only increased from 283 to 327 nm. Generally, $D_{a,PM_1}$ ranged from 255 to 381 nm, with an average of 303 nm, suggesting the relatively smaller variations $D_{a,PM_1}$ compared with $D_a$ under supersaturated conditions. While the differences in the fractions of aerosol components residing in supermicron diameters were small across different $D_a$ ranges, consistent with observed changes in aerosol size distributions, the

fractions residing in diameters larger than 2.5 µm would decrease noticeably as $D_a$ increased due to the decrease in supersaturation levels. Furthermore, under supersaturated conditions, more than 70% of the dry-state submicron aerosol mass was observed to reside in supermicron diameter ranges. It is noteworthy that aerosol mass measurements from mass spectrometry indicated that only 55% of the dry-state submicron aerosol mass resided in supermicron diameter ranges during fogs. This discrepancy can be attributed to the fact that the collection efficiency of HR-AMS is lower than

1 for diameters larger than ~600 nm (Liu et al., 2007), thus leading to a lower fraction being captured. These results provide direct evidences for the shift in cut-off size of $PM_1$ and $PM_{2.5}$ associated with aerosol hygroscopic growth and activation.

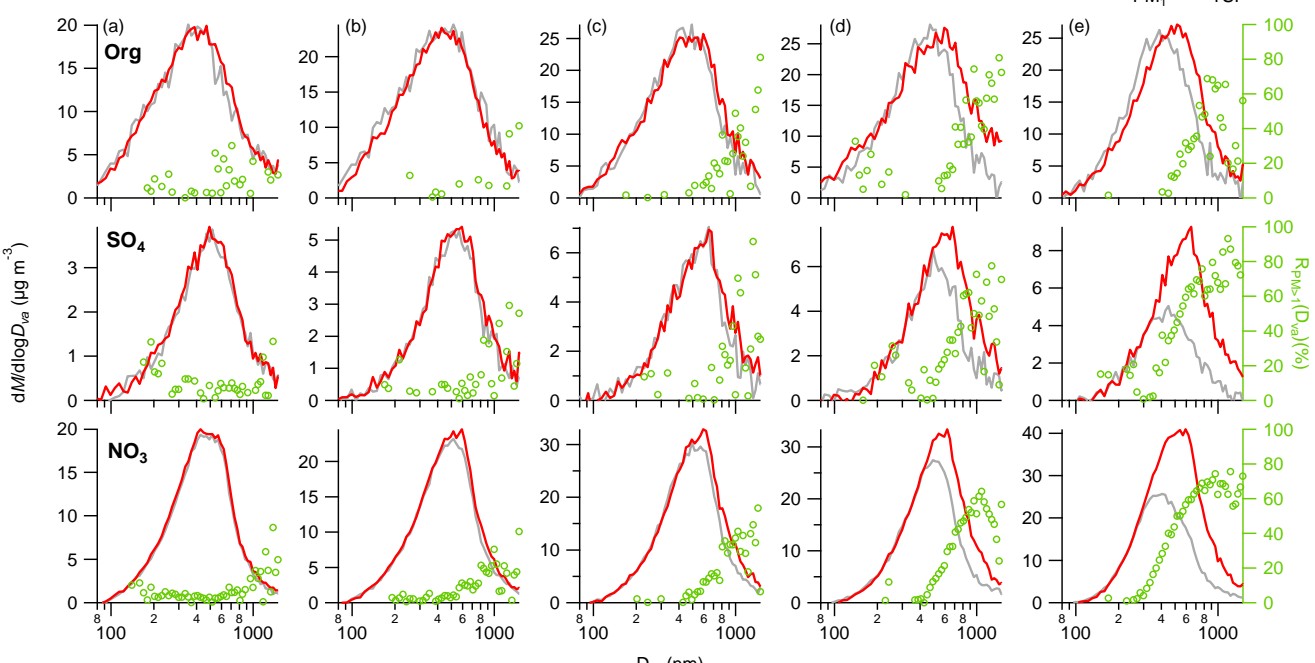

**Figure 4**. Average size distributions of OA, SO₄ and NO₃ under (a) RH=60-80%, (b) RH=80-90%, (c) RH=90-95%, (d)



RH=95-99% and (e) RH>99%. The right axis shows the corresponding average $R_{PM_{>1}}(D_{va})$.

Figure 4 shows the average size distributions of PM species under different RH ranges. The size distributions of $SO_4$ under RH < 90% were comparable between $PM_1$ and TSP, with only a few fractions of $PM_1$ $SO_4$ at $D_{va}$ < 800 nm being able to grow beyond 1 μm (Fig. 4). In comparison, the peak diameter of $PM_1$ $SO_4$ ($D_{va}$ of 520 nm) was lower than that

in TSP ($D_{va}$ of 600 nm), and the size for $PM_1$ $SO_4$ capable of growing beyond 1 μm shifted to $D_{va}$ of 400 nm under RH ranging from 95 to 99%. The variations in size distributions of $NO_3$ are generally consistent with $SO_4$ as influenced by the changes in RH. For example, the increases in $NO_3$ mass concentrations from $PM_1$ to TSP were also concentrated in particles of with $D_{va}$ 400 nm under RH = 95-99%. However, small differences between variations in size distributions of $NO_3$ and $SO_4$ were also observed. For example, the different $R_{PM_{>1}}(D_{va})$ under RH range of 95-99%, may be

attributed to their diverse mixing state across different sizes. Similar to the variations in size distributions of SIA, the increase in peak diameters of OA in TSP was faster than that in $PM_1$ as RH increased, and the TSP OA peaked at 520 nm, exceeding that of $PM_1$ (~ $D_{va}$ of 400 nm) under RH > 99%. $PM_1$ OA above ~450 nm exhibited lower mass concentrations compared to that in TSP under RH = 95-99%. Conversely, the minimal size of differences in OA concentrations between $PM_1$ and TSP were shifted to ~400 nm under RH > 99%, which was larger than that of SIA (~

$D_{va}$ of 300 nm). Figure S8 also illustrates the comparison of size distributions of PM species in $PM_{2.5}$ and TSP. Notably, the disparities between $PM_{2.5}$ and TSP are only discernible under RH>99%, which is consistent with that dry state submicron aerosols would migrate beyond 2.5 μm substantially under only supersaturated conditions. While SIA above $D_{va}$ of 400 nm showed the differences in mass concentrations between $PM_{2.5}$ and TSP, the differences in OA concentrations between $PM_{2.5}$ and TSP were concentrated at $D_{va}$ > 600 nm, suggesting the SOA with larger size

contributed more to the activation compared to POA with smaller size.

**3.3 Impact of aerosol hygroscopic growth and activation on aerosol bulk compositions**



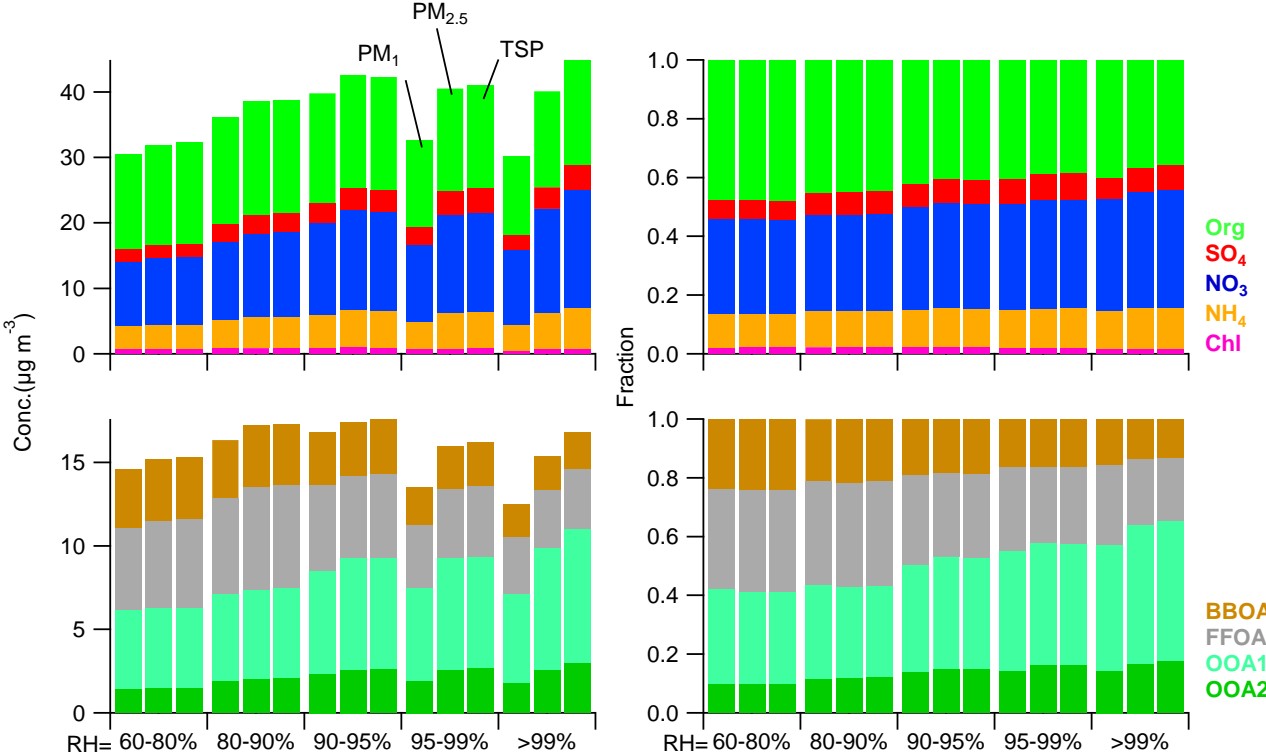

**Figure 5.** Average concentrations and contributions of PM and OA species under different size at different RH levels.

Consistent with previous studies at urban sites (Wang et al., 2015; Elser et al., 2016), the mass concentrations of PM species showed considerable increases from RH < 60% to RH > 60%, highlighting the effect of aqueous-phase processing. While comparable mass concentrations of PM species were observed under different sizes at RH < 60% due to small effects of hygroscopic growth on the cut-off size as discussed before, the increases of PM species under different sizes varies with the change of RH levels (Table S2). For example, the mass concentration of aerosols in $PM_1$ increased from 20.0 μg m$^{-3}$ under RH<80% to 39.8 μg m$^{-3}$ under RH=90-95%, while such nearly doubled mass increase was less than that in $PM_{2.5}$ and TSP. Note that the mass concentrations of NR-$PM_1$ did not continuously increase with the further increase of RH, rather decreasing to 32.6 μg m$^{-3}$ under RH=95-99% and 30.2 μg m$^{-3}$ under RH>99%, respectively. One explanation was that hygroscopic growth of $PM_1$ under high RH levels rendered some of the submicron aerosols unable to be captured by the $PM_1$ cyclone. In contrast to the negligible differences in mass concentrations between $PM_{2.5}$ and TSP under RH< 99%, significant disparities in concentrations are observed between $PM_{2.5}$ and TSP (40.1 μg m$^{-3}$ vs. 44.9 μg m$^{-3}$) under RH>99%. As discussed above, such behaviors could be attributed to the fact that the particles that grew to larger sizes due to hygroscopic water uptake and activation could not be captured by the $PM_1$ and $PM_{2.5}$ cyclones.

As expected, the hygroscopic growth and activation induced PM mass concentrations increases varied among distinct chemical species. Under RH of 95-99%, the mass concentration of $SO_4$ in $PM_{2.5}$ increased by 33% (3.6 μg m$^{-3}$) compared



to that in PM$_1$ (2.7 μg m$^{-3}$), which was larger than that for NO$_3$ (27%) and OA (18%). Additionally, OOA1 increased by 21.8% from PM$_1$ (5.5 μg m$^{-3}$) to PM$_{2.5}$ (6.7 μg m$^{-3}$) at RH = 95-99%, lower than the increase observed in OOA2 (36.8%). We also observed a 18.2% increase in BBOA concentration in PM$_{2.5}$ compared to that in PM$_1$ at RH = 95-99%, which

was larger than that for FFOA (7.9%), indicating that BBOA was more susceptible to hygroscopic growth than FFOA. Such contributions confirmed the different hygroscopic growth of POA components, consistent with previous studies (Betha et al., 2018; Kuang et al., 2024). The PM$_{2.5}$ and TSP compositions were comparable under RH < 99%, confirming the fact that aerosol activation did not occur under conditions of RH<99%. Under RH > 99%, obvious differences in compositions were detected among different sizes, with detailed mass concentrations of PM species listed in Table S2.

The particles differences that could be captured by the PM$_{2.5}$ cyclone but not by the PM$_1$ cyclone were defined as the hygroscopic growth contribution, while particles that were not captured by the PM$_{2.5}$ cyclone but measured by the TSP passage were attributed to activation contribution. SO$_4$ mass concentrations in TSP increased by 78% (1.7 μg m$^{-3}$) compared to that in PM$_1$ under RH>99%, including 1.1 μg m$^{-3}$ attributed to hygroscopic growth and 0.58 μg m$^{-3}$ to activation. Such contributions of mass concentrations from hygroscopic growth and activation to the increase in SO$_4$

were comparable to those observed for NO$_3$. OA species showed different contributions from hygroscopic growth and activation. For example, the mass concentration of OOA1 in TSP showed a 2.7 μg m$^{-3}$ increase compared to PM$_1$ under RH>99%, with 2.0 μg m$^{-3}$ attributed to hygroscopic growth and 0.7 μg m$^{-3}$ to activation. Although the absolute increase in OOA2 mass concentration was lower from PM$_1$ to TSP (0.78 μg m$^{-3}$ from hygroscopic growth and 0.38 μg m$^{-3}$ from activation), the relative contribution of activation was higher (33% for OOA2 vs. 26% for OOA1). These discrepancies

highlight differences in hygroscopic growth and activation properties of distinct SOA species likely due to diverse chemical compositions and sizes.

On average, the mass concentration of BBOA in TSP was 2.2 μg m$^{-3}$, which was larger than PM$_{2.5}$ (2.1 μg m$^{-3}$) and PM$_1$ BBOA (1.9 μg m$^{-3}$) under RH>99%, suggesting more than at least 5.8% activation for submicron BBOA. Comparatively, the mass concentrations of FFOA showed negligible differences between different sizes likely due to the insufficient

hygroscopicity of FFOA. As shown in Fig. S9, the average activation ratios of BBOA under different supersaturations were generally higher than 10% during Ep5, however, no activation of FFOA was observed. Indeed, the contributions of hygroscopic growth and activation of POA varies as the development of fog process (Figs. S10-S11). For example, the contribution of POA to the difference between PM$_{2.5}$ and TSP of OA decreased from 52.1% in S2 to 11.6% in S6 under RH>99% during Ep5. These results suggest the changes of aerosol hygroscopic growth and activation of POA containing

aerosols as the development of fog process due to the supersaturation variations and even deactivation of POA containing aerosols.



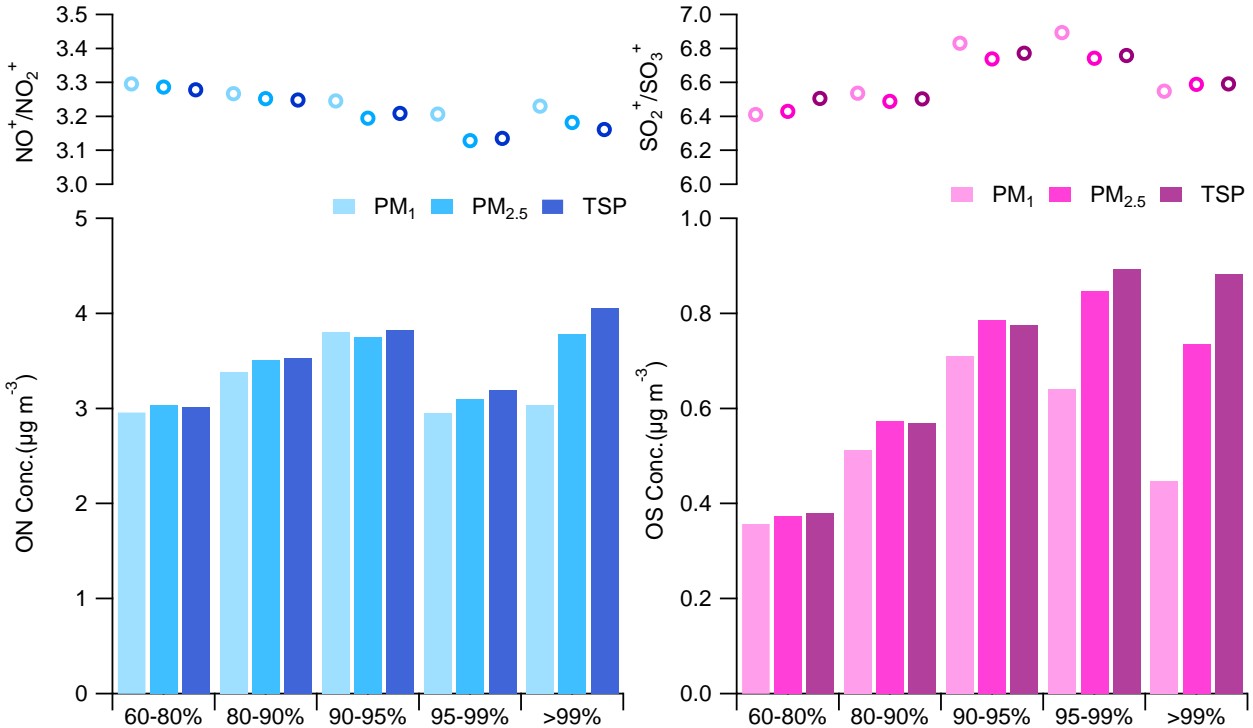

**Figure 6.** Average $NO^+/NO_2^+$, $SO_2^+/SO_3^+$ and mass concentrations of OS and lower bounds of ON under different size at different RH levels.

Organic nitrates (ON) and organosulfur compounds (OS) were estimated using the $NO_X$ method (Farmer et al., 2010) and the method proposed by Chen et al. (2019), respectively (Table S3). As depicted in Fig. 6, ON and OS revealed significant variations among different sizes across different RH levels. For example, while the mass concentrations of ON and OS were comparable under different sizes under RH < 80%, the mass concentrations of ON and OS in TSP was by 8.1% and 39.1% higher compared to that in $PM_1$, respectively, under RH ranging 95 to 99%. Under RH conditions of >99%, the mass concentration of ON in TSP showed a 4.05 μg m$^{-3}$ increase compared to $PM_1$, with 0.76 μg m$^{-3}$ attributed to hygroscopic growth and 0.26 μg m$^{-3}$ to activation. While the absolute increases in OS mass concentrations from hygroscopic growth (0.29 μg m$^{-3}$) and activation (0.14 μg m$^{-3}$) were lower than ON, the contributions of activation (25.5% vs. 32.6%) to the increased concentrations of ON were found to be lower compared to those for OS. Despite of the mass concentrations of ON and OS in TSP increases due to hygroscopic growth and activation, the $NO^+/NO_2^+$ and $SO_2^+/SO_3^+$ ratios in $PM_1$ were lower than those observed in TSP, indicating that ON and OS mass concentrations increments caused by high RH levels were of smaller magnitudes than those of $NO_3$ and $SO_4$, respectively. These findings suggest that ON and OS may have lower hygroscopicity or reside in smaller sizes than $NO_3$ and $SO_4$. Nevertheless, ON and OS were still non-negligible components of fog droplets. Considering that ON often serves as an important component of brown carbon (Laskin et al., 2015), it might have a significant implication in radiative impacts of fog and





cloud droplets by absorbing solar radiation at ultraviolet wavelengths.

## 3.4 Impact of hygroscopic growth and activation on OA mass spectra

**Figure 7**. Average high-resolution mass spectra of PM$_1$ (left panel) and TSP (middle panel) OA at (a) RH=60-80%, (b) RH=80-90%, (c) RH=90-95%, (d) RH=95-99% and (e) RH>99%. The differences in mass spectra of TSP and PM$_1$ OA are shown on right panel.

While the mass spectra of OA were comparable under RH < 80% across different sizes (Fig.7) due to similar chemical compositions, the variations in ion categories under different sizes as a function of RH revealed significant differences. For example, the C$_x$H$_y^+$ of PM$_1$/TSP ratio exhibited substantial increases as the increases of RH, with values larger than 1 at RH > 90% (Fig. S12). This suggests a decreased contribution of C$_x$H$_y^+$ in TSP at high RH levels, implying less hygroscopic growth potential of POA compared to SOA. The C$_x$H$_y$O$_1^+$ of PM$_1$/TSP ratios were below 1 under RH < 95% across all m/z's, but exceeded 1 at m/z > 60 under RH > 95%. The behavior of C$_x$H$_y$O$_2^+$ did not exhibit a clear changing pattern with RH. These findings underscore the differences in hygroscopic growth and activations of various molecular compositions, yet it remains a challenge to identify them by HR-AMS. Furthermore, comparisons of typical ion mass





concentrations between different sizes under different RH levels revealed distinct responses of chemical compounds to
RH changes. $CO_2^+$ and $CHO^+$, which are associated with SOA and aqueous processes (Zhao et al., 2019), exhibited a
20.0% increase in TSP compared to that in $PM_1$ under RH = 95-99%, respectively. Additionally, the N/C in $PM_1$ was
lower than that in TSP under RH>90%, and a significant $CH_4N^+$ signal was observed, implying the contribution of N-
containing compounds to hygroscopic growth and activation. These results indicate different responses of chemical
compounds to RH level changes, with almost all species showing significant increases in the TSP compared to that in
$PM_1$ at RH > 95%.

While similar O/C ratios were generally observed among different sizes at RH<95%, the O/C in TSP (0.64) was higher
than that in $PM_1$ (0.59) under RH > 99%, likely due to the change in chemical compositions under different sizes due to
activation. As shown in Fig.7, the dominant contribution to the difference between $PM_1$ and TSP was attributed to $CO_2^+$
and $CO^+$, followed by $CHO^+$ and $C_2H_3O^+$. One possible explanation is that particles with relatively higher O/C possibly
exhibit higher hygroscopicity and are more susceptible to hygroscopic growth, rendering part of them unable to be
captured by the $PM_1$ cyclone under RH > 99% (Kuang et al., 2020). This interpretation is supported by the observation
that SOA contributed more to the increase in TSP OA concentrations than POA at supersaturated conditions. The $C_xH_y^+$
ion categories also contributed in part to the disparity between $PM_1$ and TSP under RH>99%, likely due to the activation
of POA.

## 4 Conclusions

This study comprehensively characterized the composition, sources, and size distributions of submicron aerosols
utilizing an advanced fog-aerosol sampling system at a rural site in NCP. Highly hygroscopic aerosol components, such
as inorganic nitrate and sulfate, exhibited significant variations under different sizes, particularly under higher RH
conditions. For instance, under RH=95-99%, the mass concentration of $PM_{2.5}$ $SO_4$ increased by 33% compared to $PM_1$,
surpassing the increases observed for $NO_3$ and OA. Conversely, the $PM_{2.5}$ and TSP compositions were similar to each
other under RH<99%. Our results indicate that aerosols ranging from 1 to 2.5 μm experience relatively facile hygroscopic
growth but face challenges in growing beyond 2.5 μm in size. Furthermore, we observed increased ON and OS mass
concentrations in TSP compared to those in $PM_1$ despite decreases in $NO^+/NO_2^+$ and $SO_2^+/SO_3^+$. During supersaturated
fog conditions, over 70% of dry state submicron aerosol mass was found to reside in supermicron diameter ranges. The
critical diameter for growth beyond the supermicron range ranged from 255 to 381 nm, with an average of 303 nm, while
the critical activation diameter ranged from 362 to 664 nm, averaging 487 nm. These results highlight significant impacts
of aerosol hygroscopic growth and activation on aerosol sampling under high RH conditions.
The study also provided insights into the activation characteristics of SOA and POA factors during fog periods, shedding
light on the aqueous processing of POA and SOA conversion. For instance, the potential aqueous conversion of BBOA
in fog and cloud droplets was indicated by its activation abilities to form fog droplets, which would depend on
supersaturations and other environmental factors. However, the efficiency of aqueous conversion for FFOA in fogs
appeared to be low, as only small portions of submicron FFOA could grow beyond 1 μm. Furthermore, distinct ion



categories including $C_xH_y^+$, $C_xH_yO_1^+$, $C_xH_yO_2^+$, $C_xH_yN_p^+$ in $PM_1$/TSP ratios under different RH levels suggested varying hygroscopic growth and activation abilities as well as diverse chemical compounds. However, detailed mechanisms require further comprehensive investigations. Overall, these results demonstrate significantly different cloud and fog processing behaviors between primary and secondary aerosols.

**Data availability**. Data will be made available on request.

**Author contribution**. WeX, YK and YS designed the research. WeX, YK, WaX, BL, XZ, JT and HQ conducted the measurements. WeX, YK, WaX and YS analysed the data. WaX, ZZ, LL and YS reviewed and commented on the paper. WeX and YK wrote the paper.

**Competing interests**. The authors declare that they have no conflict of interest.

**Acknowledgements.** This work was supported by the National Natural Science Foundation of China (No. 42377101, 42175083)

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
