# Peer review of "Hygroscopic Growth and Activation Changed Submicron Aerosol Composition and Properties in North China Plain"

_EGUsphere, 2024_

## Referee Comment (RC2)

This study presents a thorough discussion on how the hygroscopic growth influences the sampling, aerosol size and composition under high RH condition. The datasets and analysis are robust and provide valuable insights for high-RH atmosphere. I have some suggestions.

1. During the fog events, atmospheric RH reaches ~100% for nearly harf day. At our city, if this happens, usually it's accompanied with precipitation. Is there any precipitation during the observation periods? How frequent is this foggy condition happens at this site and other NCP cities?

2. RH usually has a diurnal variation, which is higher at night and lower at day. When studying the composition and source reliance on RH, the diurnal variation of RH is mixed with the diurnal variations of different composition and PMF factors. How to understand this influence on Figure 5-7?

3. The hygroscopic growth will change the particle diameter thus shifting the cut-off size of the impactors. It is better to quantify how much underestimation it will cause to PM2.5 and PM1 mass under different RH. This will provide reference for other studies.

4. How about moving the nafion dryer in front of the cyclone impactors? Will this solve the hygroscopic influence on cut-off sizes?

---

## Author Comment (AC1)

**Response to Reviewer #1**

*General comments:*

*This study used an advanced aerosol-fog sampling system to directly investigate the impacts of aerosol hygroscopic growth and activation processes on physical and chemical properties of submicron aerosols which is rarely done before. Presented results revealed significant shifts in aerosol size distributions and composition under both subsaturated and supersaturated high RH conditions, with more than 70% submicron aerosol migrated to supermicron ranges in foggy conditions, and over 25% of aerosol mass with dry diameters below 1 μm resided in supermicron ranges under subsaturated high RH (> 90%), shedding new insights into aerosol sampling and aerosol physical properties in foggy/cloudy and high RH conditions. Moreover, simultaneous measurements of aerosol chemical compositions shedding new insights into hygroscopic and activation properties of different aerosols types, which is important implications in both aerosol/cloud aqueous chemistry and cloud microphysics. In general, I think this manuscript is well organized and presented direct measurements of aerosol hygroscopic and activation processes with the view of aerosol size and chemistry which is very useful for aerosol and fog/cloud community,therefore suggest acceptance after minor revisions.*

We thank the reviewer's positive comments.

*Minor comments:*

*Emphasize the broader implications of the research findings for aerosol/cloud science and atmospheric chemistry in the abstract*
 Corrected.
Following the reviewer's comments, we added
"Overall, our study highlights remarkably different cloud and fog processing behaviors between primary and secondary aerosols, which would benefit a better understanding of aerosol-cloud interactions under distinct atmospheric conditions."

*Explicitly state the research objectives or questions towards the end of the introduction to guide readers on what to expect from the study. This helps to focus the reader's attention and provides a roadmap for the rest of the manuscript. Current version discussed too much on aerosol chemical compositions, however, discussions about current understandings of how aerosol hygroscopic growth and activation impacts on aerosol submicron aerosol compositions is relatively few.*

We thank the reviewer's comments. Indeed, we have expanded our discussion in the introduction to further explore how RH impacts aerosol chemical compositions, as only a limited number of studies have directly investigated the effects of aerosol hygroscopic growth and activation on aerosol size or chemical compositions.

To better clarify this point, the following sentences are added in the first paragraph: "Most current studies primarily focus on the impact of RH on aerosol chemical compositions, with only a limited number of studies directly quantifying how aerosol hygroscopic growth and activation affect size and/or chemical compositions of aerosol"

---

## Author Comment (AC2)

**Response to Reviewer #2**

*This study presents a thorough discussion on how the hygroscopic growth influences the sampling, aerosol size and composition under high RH condition. The datasets and analysis are robust and provide valuable insights for high-RH atmosphere. I have some suggestions.*

We thank the reviewer's comments. We have revised the manuscript accordingly.

*1. During the fog events, atmospheric RH reaches ~100% for nearly half day. At our city, if this happens, usually it's accompanied with precipitation. Is there any precipitation during the observation periods? How frequent is this foggy condition happens at this site and other NCP cities?*

Thanks for the reviewer's comment. Fog accompanied with precipitation typically occurs in mountainous regions or at land-sea boundaries where increased moisture can lead to saturation. In contrast, fog formation in the North China Plain (NCP), especially during autumn and winter, may be accompanied by a significant decrease in temperature under stagnant conditions due to low moisture levels. Fu et al. (2014) summarized the frequency and trends of fogs in the NCP over the past 30 years. In this study, the observed fogs are primarily radiation fog caused by substantial nighttime temperature decreases under calm conditions (i.e., low wind speeds) (Kuang et al., 2024), with no associated precipitation events during the observation period. As this study does not focus on precipitation, it is not discussed in this manuscript."

*2. RH usually has a diurnal variation, which is higher at night and lower at day. When studying the composition and source reliance on RH, the diurnal variation of RH is mixed with the diurnal variations of different composition and PMF factors. How to understand this influence on Figure 5-7?*

[Figure]

Fig. R1. Diurnal frequencies of different RH levels.

[Figure]

Fig. R2. The concentrations PM and OA species in PM$_1$(left axis) and weighted by the diurnal RH frequency only (right axis) under different RH levels.

We thank the reviewer's comments. We do acknowledge the potential impact of diurnal variations of RH on the discussion of the composition and source reliance on RH. As depicted in Fig. R1, there is a higher frequency during 0-4 a.m. under RH>99% and a higher frequency during 8-12 a.m. under RH<60%. This diurnal variation may introduce uncertainties regarding the variation in physicochemical properties under changing RH levels. However, upon further comparison of PM and OA species concentrations weighted by diurnal RH frequency only ($=\sum_i F_i \times C_i$, where $F_i$ and $C_i$ are the frequency and concentrations for each hour), we observed negligible variations in PM and OA species concentrations under different levels of RH (Fig. R2). These findings differ significantly from those observed in PM$_1$. Therefore, the impact of diurnal variations in RH is not considered as the primary influencing factor.

*3. The hygroscopic growth will change the particle diameter thus shifting the cut-off size of the impactors. It is better to quantify how much underestimation it will cause to PM$_{2.5}$ and PM$_1$ mass under different RH. This will provide reference for other studies.* Following the reviewer's comments, we added the differences between PM$_{2.5}$ and PM$_1$ in mass concentrations and the contribution of PM$_1$ to PM$_{2.5}$ in Table S1.

Table S1. A summary of differences of mass concentrations between TSP, PM$_{2.5}$ and PM$_1$, and the fraction of PM$_1$ in PM$_{2.5}$ and TSP of PM Species and OA factors under different RH levels.

|  | <60% | 60-80% | 80%-90% | 90-95% | 95-99% | >99% |
|---|---|---|---|---|---|---|
| PM$_{2.5}$-PM$_1$ (µg m$^{-3}$) |  |  |  |  |  |  |
| Org | 0.4 | 0.7 | 1.0 | 0.5 | 2.4 | 2.5 |
| NO$_3$ | 0.2 | 0.5 | 0.8 | 1.4 | 3.3 | 4.4 |
| SO$_4$ | 0.0 | 0.1 | 0.3 | 0.4 | 0.9 | 1.1 |
| NH$_4$ | 0.1 | 0.2 | 0.3 | 0.6 | 1.2 | 1.6 |
| Chl | 0.1 | 0.1 | 0.1 | 0.1 | 0.1 | 0.1 |
| FFOA | 0.2 | 0.3 | 0.3 | -0.3 | 0.3 | 0.1 |
| BBOA | 0.1 | 0.2 | 0.2 | 0.0 | 0.4 | 0.2 |
| OOA1 | 0.0 | 0.1 | 0.2 | 0.6 | 1.2 | 2.0 |

| | | | | | | |
|---|---|---|---|---|---|---|
| OOA2 | 0.0 | 0.1 | 0.1 | 0.3 | 0.7 | 0.8 |
| **PM$_1$/PM$_{2.5}$(%)** | | | | | | |
| Org | 93.8 | 95.4 | 94.3 | 97.1 | 84.7 | 83.0 |
| NO$_3$ | 96.2 | 95.1 | 93.7 | 90.8 | 78.0 | 72.3 |
| SO$_4$ | 100.0 | 95.2 | 90.0 | 88.6 | 75.0 | 66.7 |
| NH$_4$ | 94.7 | 94.6 | 93.6 | 89.5 | 78.2 | 71.4 |
| Chl | 66.7 | 85.7 | 88.9 | 88.9 | 85.7 | 83.3 |
| FFOA | 86.7 | 94.3 | 95.1 | 106.1 | 92.7 | 97.1 |
| BBOA | 88.9 | 94.6 | 94.6 | 100.0 | 84.6 | 90.5 |
| OOA1 | 100.0 | 97.9 | 96.3 | 91.0 | 82.1 | 72.6 |
| OOA2 | 100.0 | 93.3 | 95.0 | 88.5 | 73.1 | 69.2 |
| **TSP-PM$_1$ (µg m$^{-3}$)** | | | | | | |
| Org | 0.7 | 1.0 | 1.0 | 0.5 | 2.5 | 3.8 |
| NO$_3$ | 0.2 | 0.5 | 0.9 | 1.2 | 3.4 | 6.4 |
| SO$_4$ | 0.0 | 0.1 | 0.3 | 0.3 | 1.1 | 1.7 |
| NH$_4$ | 0.1 | 0.2 | 0.4 | 0.5 | 1.3 | 2.4 |
| Chl | 0.1 | 0.3 | 0.1 | 0.1 | 0.2 | 0.2 |
| FFOA | 0.3 | 0.3 | 0.4 | -0.2 | 0.5 | 0.2 |
| BBOA | 0.3 | 0.2 | 0.2 | 0.1 | 0.4 | 0.3 |
| OOA1 | 0.1 | 0.1 | 0.2 | 0.6 | 1.2 | 2.7 |
| OOA2 | 0.0 | 0.1 | 0.2 | 0.3 | 0.8 | 1.1 |
| **PM$_1$/TSP(%)** | | | | | | |
| Org | 89.6 | 93.5 | 94.3 | 97.1 | 84.2 | 76.3 |
| NO$_3$ | 96.2 | 95.1 | 93.0 | 92.1 | 77.5 | 64.2 |
| SO$_4$ | 100.0 | 95.2 | 90.0 | 91.2 | 71.1 | 56.4 |
| NH$_4$ | 94.7 | 94.6 | 91.7 | 91.1 | 76.8 | 62.5 |
| Chl | 66.7 | 66.7 | 88.9 | 88.9 | 75.0 | 71.4 |
| FFOA | 81.3 | 94.3 | 93.5 | 104.0 | 88.4 | 94.4 |
| BBOA | 72.7 | 94.6 | 94.6 | 97.0 | 84.6 | 86.4 |
| OOA1 | 97.0 | 97.9 | 96.3 | 91.0 | 82.1 | 66.3 |
| OOA2 | 100.0 | 93.3 | 90.5 | 88.5 | 70.4 | 62.1 |

*4. How about moving the nafion dryer in front of the cyclone impactors? Will this solve the hygroscopic influence on cut-off sizes*

We thank the reviewer's comments. The particles affected by hygroscopic growth and activation were collected by three different cyclone impactors and then dried by nafion dryer. And hence the physicochemical properties of these particles were characterized by following instruments. Indeed, by drying the particles before collection via impactors, regardless of high RH, saturated foggy or cloudy conditions, the aerosols would revert to their size under dry conditions. As a result, the three impactors will collect almost identical particles due to the small proportion of dry aerosols in particle sizes larger than 1 µm, thus unable to reflect the hygroscopic and activation characteristics of particles in ambient air. Therefore, in order to eliminate aerosol hygroscopic growth and activation during sampling, it is advisable to first dry the

samples and then select different size ranges using impactors.

References

Fu, G. Q., Xu, W. Y., Yang, R. F., Li, J. B., and Zhao, C. S.: The distribution and trends of fog and haze in the North China Plain over the past 30 years, Atmos. Chem. Phys., 14, 11949-11958, 10.5194/acp-14-11949-2014, 2014.

Kuang, Y., Xu, W., Tao, J., Luo, B., Liu, L., Xu, H., Xu, W., Xue, B., Zhai, M., Liu, P., and Sun, Y.: Divergent Impacts of Biomass Burning and Fossil Fuel Combustion Aerosols on Fog-Cloud Microphysics and Chemistry: Novel Insights From Advanced Aerosol-Fog Sampling, Geophysical Research Letters, 51, e2023GL107147, https://doi.org/10.1029/2023GL107147, 2024.